# The transformation of sensory to perceptual braille letter representations in the visually deprived brain

**Marleen Haupt[1]\*[†], Monika Graumann[1,2][†], Santani Teng[3], Carina Kaltenbach[1], Radoslaw Cichy[1,2,4]\***

[1]Department of Education and Psychology, Freie Universität Berlin, Berlin, Germany; [2]Berlin School of Mind and Brain, Faculty of Philosophy, Humboldt-Universität zu Berlin, Berlin, Germany; [3]Smith-Kettlewell Eye Research Institute, San Francisco, United States; [4]Bernstein Center for Computational Neuroscience Berlin, Berlin, Germany

**\*For correspondence:**
marleen.haupt@gmail.com (MH);
rmcichy@gmail.com (RC)

[†]These authors contributed equally to this work

**Competing interest:** The authors declare that no competing interests exist.

## eLife Assessment

This **valuable** study investigates the brain representations of Braille letters in blind participants and provides evidence using EEG and fMRI that the decoding of letter identity across the reading hand takes place in the visual cortex. The evidence supporting the claims of the authors is **convincing** and the work will be of interest to neuroscientists working on brain plasticity.

**Abstract** Experience-based plasticity of the human cortex mediates the influence of individual experience on cognition and behavior. The complete loss of a sensory modality is among the most extreme such experiences. Investigating such a selective, yet extreme change in experience allows for the characterization of experience-based plasticity at its boundaries. Here, we investigated information processing in individuals who lost vision at birth or early in life by probing the processing of braille letter information. We characterized the transformation of braille letter information from sensory representations depending on the reading hand to perceptual representations that are independent of the reading hand. Using a multivariate analysis framework in combination with functional magnetic resonance imaging (fMRI), electroencephalography (EEG), and behavioral assessment, we tracked cortical braille representations in space and time, and probed their behavioral relevance. We located sensory representations in tactile processing areas and perceptual representations in sighted reading areas, with the lateral occipital complex as a connecting 'hinge' region. This elucidates the plasticity of the visually deprived brain in terms of information processing. Regarding information processing in time, we found that sensory representations emerge before perceptual representations. This indicates that even extreme cases of brain plasticity adhere to a common temporal scheme in the progression from sensory to perceptual transformations. Ascertaining behavioral relevance through perceived similarity ratings, we found that perceptual representations in sighted reading areas, but not sensory representations in tactile processing areas are suitably formatted to guide behavior. Together, our results reveal a nuanced picture of both the potentials and limits of experience-dependent plasticity in the visually deprived brain.

## Introduction

Human brains vary due to individual experiences. This so-called experience-based plasticity of the human cortex mediates cognitive and behavioral adaptation to changes in the environment

(*Pascual-Leone et al., 2005*). Typically, plasticity reflects learning from species-typical experiences. However, plasticity also results from species-atypical changes to experience like the loss of a sensory modality.

Sensory loss constitutes a selective, yet large-scale change in experience that offers a unique experimental opportunity to study cortical plasticity at its boundaries (*Ricciardi et al., 2020*). One deeply investigated case of sensory loss is blindness, that is, the lack of visual input to the brain. Previous research has shown that cortical structures most strongly activated by visual input in sighted brains are activated by a plethora of other cognitive functions in visually deprived brains (*Bedny, 2017*), including braille reading (*Büchel et al., 1998*; *Burton et al., 2012*; *Rączy et al., 2019*; *Reich et al., 2011*; *Sadato et al., 1998*; *Uhl et al., 1991*; *Sadato, 1996*). However, overlapping functional responses alone cannot inform us about the nature of the observed activations, that is, what kind of information they represent and thus what role they play in cognitive processing.

To elucidate the nature of information processing in the visually deprived brain, we investigate the tactile braille system in individuals who lost vision at birth or early in life (hereafter blind participants). Braille readers commonly use both hands, requiring their brain to transform sensory tactile input into a hand-independent perceptual format. We made use of this practical everyday requirement to experimentally characterize the transformation of sensory to perceptual braille letter representations. We operationalize sensory braille letter representations as representations coding information specific to the hand that was reading (hand-dependent). In contrast, we operationalize perceptual braille letter representations as representations coding information independent of which hand was reading (hand-independent).

Combining this operationalization with fMRI and EEG in a multivariate analysis framework (*Cichy et al., 2013*; *Cichy et al., 2011*; *Carlson et al., 2011a*; *Isik et al., 2014*), we determine the cortical location and temporal emergence of sensory and perceptual representations. Lastly, to ascertain the functional role of the identified representations, we relate them to behavioral similarity ratings (*Cichy et al., 2019*; *Bankson et al., 2018*; *Mur et al., 2013*; *Charest et al., 2014*).

## Results

We recorded fMRI (*N* = 15) and EEG (*N* = 11) data while blind participants (see *Supplementary file 1*) read braille letters with their left or right index finger. We delivered the braille stimuli using single piezo-electric refreshable cells. This allowed participants to read braille letters without moving their finger, thus avoiding finger motion artifacts in the brain signal and analyses.

We used a common experimental paradigm for fMRI and EEG that was adapted to the specifics of each imaging modality. The common stimulus set consisted of 10 different braille letters (*Figure 1A*). Eight letters entered the main analysis. Two letters (E and O) served as vigilance targets to which participants responded with their foot; these trials were excluded from all analyses. The stimuli were presented in random order, with each trial consisting of a 500-ms stimulus presentation to either the right or left hand. In fMRI, all trials were followed by an inter-stimulus interval (ISI) of 2500 ms to account for the sluggishness of the BOLD response (*Figure 1B*). In EEG, standard trials had an ISI of 500 ms while catch trials had an ISI of 1100 ms in order to avoid movement contaminations.

The common experimental paradigm for fMRI and EEG allowed us to use an equivalent multivariate classification scheme to track the transformation of sensory to perceptual representations. We assessed fMRI voxel patterns to reveal where sensory braille letter representations are located in the cortex. Likewise, we assessed EEG electrode patterns to reveal the temporal dynamics of braille letter representations (*Figure 1C*).

We operationalized sensory versus perceptual representations as hand-dependent versus hand-independent braille letter representations, respectively. To measure perceptual representations, we trained classifiers on brain data recorded during stimulation of one hand and tested the classifiers on data recorded during the stimulation of the other hand (*Figure 1D*). We refer to this analysis as across-hand classification. It reveals perceptual braille letter representations that are independent of the specific hand being stimulated.

To assess sensory representations, we used a two-step procedure. In a first step, we trained and tested classifiers on brain data recorded during stimulation of the same hand (*Figure 1D*). We refer to this analysis as within-hand classification. It reveals both sensory and perceptual braille letter

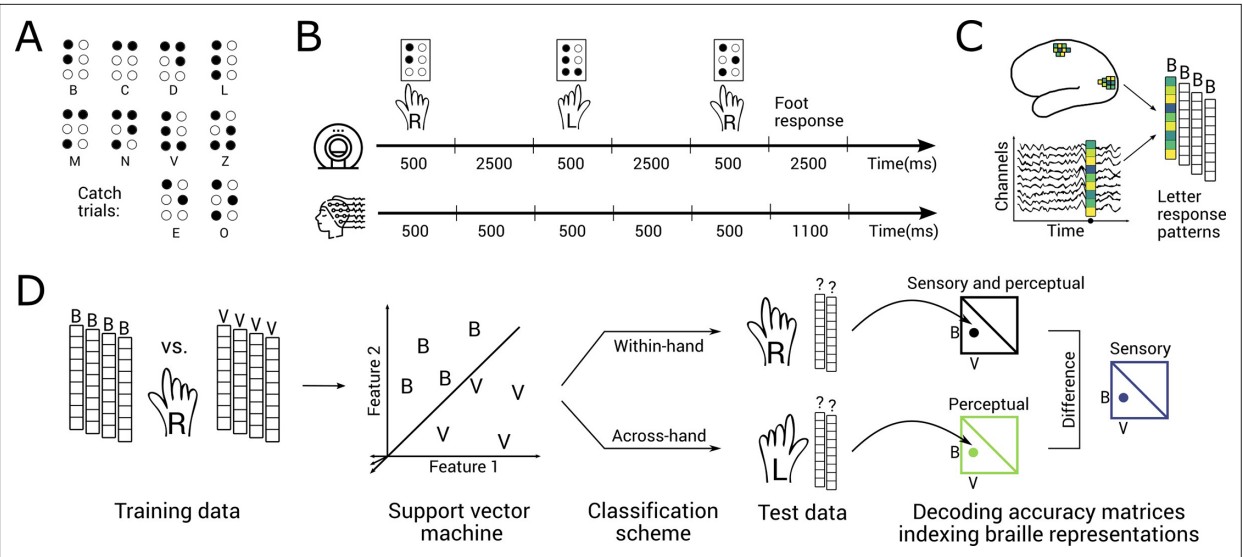

**Figure 1.** Stimuli, experimental design, pattern extraction, and multivariate analysis framework. (**A**) We presented eight braille letters (B, C, D, L, M, N, V, and Z) to participants on braille cells. Two additional letters (E and O) served as catch trials and were excluded from all analyses. (**B**) Top: During the fMRI session, braille letters were presented for 500 ms with an inter-stimulus interval (ISI) of 2500 ms. Participants were instructed to respond to catch trials by pressing a button with their foot. Bottom: During a separate EEG session, braille letters were presented for 500 ms. The ISI following regular trials lasted 500 ms, the ISI following catch trials lasted 1100 ms to avoid movement confounds. Participants were instructed to respond to catch trials by pressing a foot pedal. (**C**) In fMRI, we extracted voxel-wise activations for every region of interest (ROI). In EEG, we extracted channel-wise activations for every time point. In both cases, this resulted in one response vector per letter and per experimental run. (**D**) For both fMRI and EEG, we divided pattern vectors into training (four pseudo-runs) and test (one pseudo-run) sets. For every pair of braille letters (e.g., B and V), we trained a support vector machine (SVM) to classify between pattern vectors related to the presentation of both letters read with the same hand. We then tested the SVM on the left-out pattern vectors related to the presentation of the same two letters read with the same hand (within-hand classification) or with the other hand (across-hand classification). The resulting pairwise decoding accuracies were aggregated in a decoding accuracy matrix that is symmetric along the diagonal, with the diagonal itself being undefined. We interpret the within-hand matrix (black) as a measure of sensory and perceptual braille letter representations. We interpret the across-hand matrix (green) as a measure of perceptual braille letter representations. We derive the measure of sensory braille letter representations (blue) by subtracting one matrix from the other.

The online version of this article includes the following figure supplement(s) for figure 1:

**Figure supplement 1.** Confusion matrices for region of interest (ROI) decoding within-hand.

**Figure supplement 2.** Confusion matrices for region of interest (ROI) decoding across-hand.

**Figure supplement 3.** Confusion matrices for EEG time decoding at indicated time points within-hand.

**Figure supplement 4.** Confusion matrices for EEG time decoding at indicated time points across-hand.

representations. Thus, to further isolate sensory representations from perceptual representations, in a second step, we subtracted across-hand classification from within-hand classification results.

## Spatial dynamics of braille letter representations

We started the analyses by determining the locations of sensory and perceptual braille letter representations in the visually deprived brain using fMRI. We focused our investigation on two sets of cortical regions based on previous literature: tactile processing areas and sighted reading areas (*Figure 2A*). Given the tactile nature of braille, we expected braille letters to be represented in the tactile processing stream encompassing somatosensory cortices (S1 and S2), intra-parietal cortex (IPS), and insula (*Dijkerman and de Haan, 2007*). Given that visual reading information is processed in a ventral processing stream (*DiCarlo and Cox, 2007*; *DiCarlo et al., 2012*; *Goodale and Milner, 1992*), and that braille reading has been observed to elicit activations along those nodes (*Büchel et al., 1998*; *Burton et al., 2012*; *Rączy et al., 2019*; *Reich et al., 2011*; *Sadato et al., 1998*), we investigated the sighted processing stream ranging from early visual cortex (EVC) (*Dehaene et al., 2005*) over V4 (*Cohen et al., 2000*) and the lateral occipital complex (LOC) (*Neudorf et al., 2022*) to the letter form area (LFA) (*Thesen et al., 2012*) and visual word form area (VWFA) (*Cohen et al., 2000*; *Neudorf et al., 2022*; *McCandliss et al., 2003*; *Dehaene and Cohen, 2011*).

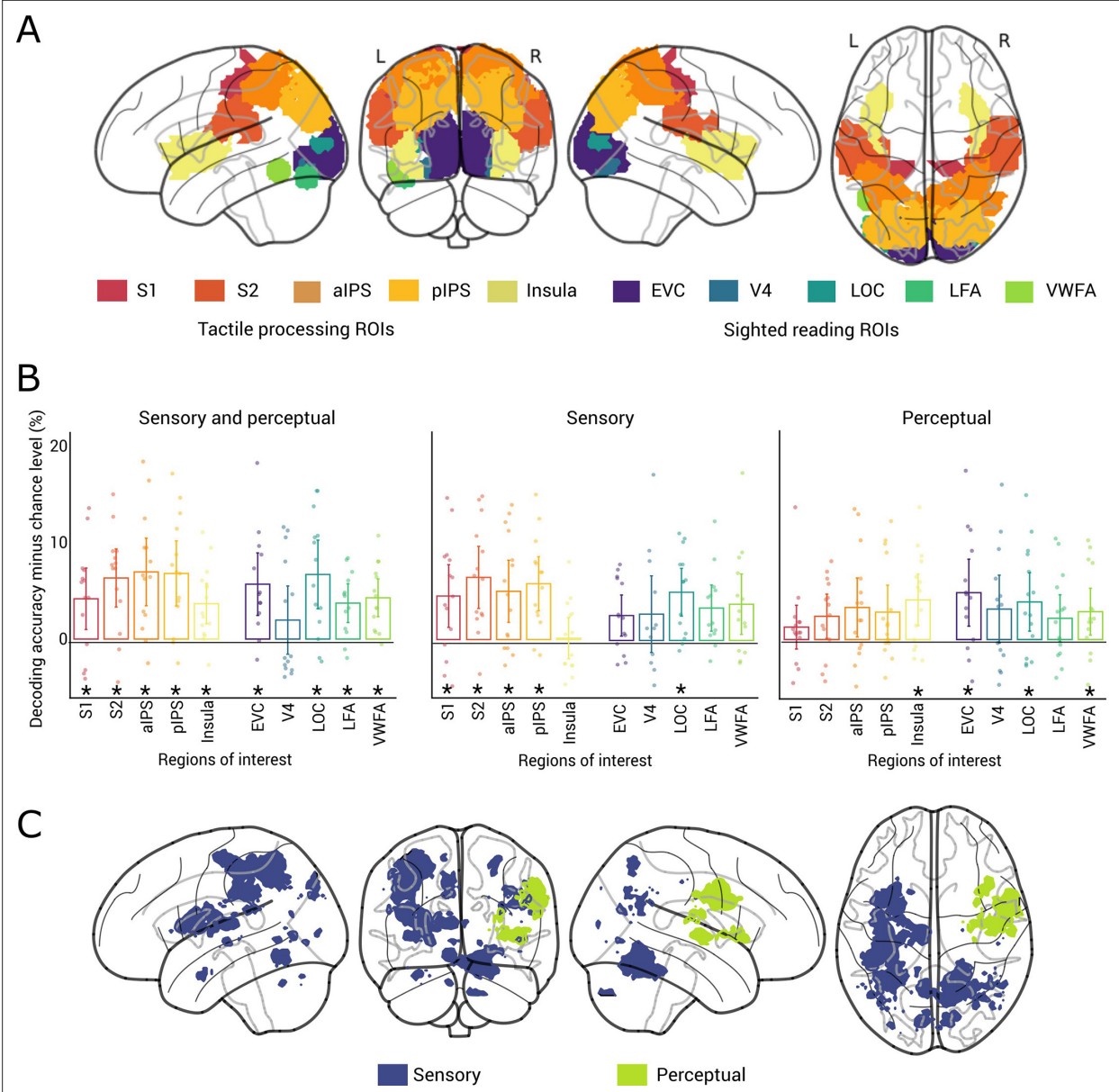

**Figure 2.** Spatial dynamics of braille letter representations. (**A**) Rendering of regions of interests associated with tactile processing (S1: primary somatosensory cortex, S2: secondary somatosensory cortex, aIPS: anterior intra-parietal sulcus, pIPS: posterior intra-parietal sulcus, insula) and sighted reading (EVC: early visual cortex, V4: visual area 4, LOC: lateral occipital complex, LFA: letter form area, VWFA: visual word form area). (**B**) Sensory and perceptual (left), sensory (middle), and perceptual (right) braille letters representations in tactile processing and sighted reading regions of interest (ROIs) $N = 15$, two-tailed Wilcoxon signed-rank test, $p < 0.05$, false discovery rate (FDR) corrected; stars below bars indicate significance above chance. Error bars represent 95% confidence intervals. Dots represent single subject data. (**C**) fMRI searchlight results for sensory (blue) and perceptual (green) braille letter representations ($N = 15$, height threshold $p < 0.001$, cluster-level family-wise error (FWE) corrected $p < 0.05$, colored voxels indicate significance). Results for combined sensory and perceptual representations are in *Figure 2—figure supplement 1*.

The online version of this article includes the following figure supplement(s) for figure 2:

**Figure supplement 1.** fMRI searchlight results for within-hand braille letter classification.

We hypothesized (H1) that sensory braille letter information is represented in tactile processing areas (H1.1), while perceptual braille letter representations are located in sighted reading areas (H1.2). To test H1, we conducted within- and across-hand classification of braille letters in the above-mentioned areas for tactile processing and sighted reading (*Figure 2A*) in a region-of-interest (ROI) analysis.

We found that within-hand classification of braille letters was significantly above chance in regions associated with both tactile processing (S1, S2, anterior intra-parietal sulcus [aIPS], posterior intra-parietal sulcus [pIPS], and insula) and sighted reading (EVC, LOC, LFA, and VWFA) (*Figure 2B*, left; *N* = 15, one-tailed Wilcoxon signed-rank test, p < 0.05, false discovery rate [FDR] corrected). As expected, this reveals both tactile and sighted reading areas as potential candidate regions housing sensory and perceptual braille letter representations.

To pinpoint the loci of sensory representations we subtracted the results of the across-hand classification from the within-hand classification. We found the difference to be significant in tactile processing areas (S1, S2, aIPS, and pIPS) and in LOC, but not elsewhere (*Figure 2B*, middle). This confirms H1.1 in that sensory braille letter representations are located in tactile areas.

To determine the location of perceptual representations, we assessed the results of across-hand classification of braille letters. We found significant information (*Figure 2B*, right) only in sighted reading areas (EVC, LOC, and VWFA), with the exception of the insula. This confirms H1.2 in that perceptual braille letter representations emerge predominantly in sighted reading areas. The surprising finding of the insula can possibly be explained by the insula's heterogeneous functions (*Uddin et al., 2017*) beyond tactile processing.

Notably, LOC is the only region that contained both sensory and perceptual braille letter representations. This suggests a 'hinge' function in the transformation from sensory to perceptual braille letter representations.

To ascertain whether other areas beyond our hypothesized ROIs contained braille letter representations, we conducted a spatially unbiased fMRI searchlight classification analysis. The results confirmed that braille letter representations are located in the assessed ROIs without revealing any

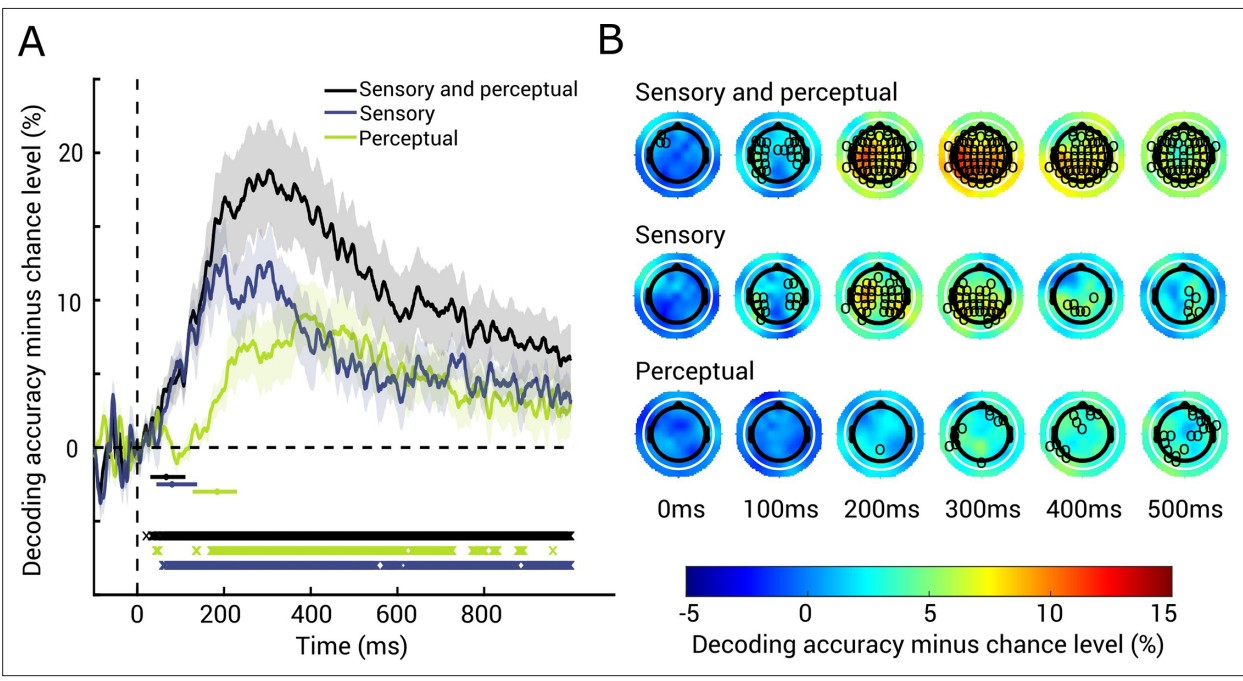

**Figure 3.** Temporal dynamics of braille letter representations. (**A**) EEG results for sensory and perceptual (black), sensory (blue), and perceptual (green) braille letter representations in time. Shaded areas around curves indicate standard error of the mean. Significance onsets and their 95% confidence intervals are indicated by dots and horizontal lines below curves (color-coded as curves, *N* = 11, 1000 bootstraps). Significant time points are indicated by x below curves (*N* = 11, one-tailed Wilcoxon signed-rank test, p < 0.05, false discovery rate [FDR] corrected). (**B**) EEG searchlight results for sensory and perceptual (top), sensory (middle), and perceptual (bottom) braille letter representations in EEG channel space (sampled down to 10 ms resolution) in 100-ms intervals. Significant electrodes are marked with black circles (*N* = 11, one-tailed Wilcoxon signed-rank test, p <0 .05, FDR corrected across electrodes and time points).

The online version of this article includes the following figure supplement(s) for figure 3:

**Figure supplement 1.** EEG time decoding results with empirical baseline (see *Figure 3A* for comparison).

**Figure supplement 2.** EEG results for decoding of stimulated hand in time.

**Figure supplement 3.** EEG time decoding results when only using occipital and parieto-occipital electrodes.

additional regions (*Figure 2C*, *N* = 15, height threshold p < 0.001, cluster-level family-wise error [FWE] corrected p < 0.05).

Together, our fMRI results revealed sensory braille letter representations in tactile processing areas and perceptual braille letter representations in sighted reading areas, with LOC serving as a 'hinge' region between them.

## Temporal dynamics of braille letter representations

We next determined the temporal dynamics with which braille letter representations emerge using EEG.

We hypothesized (H2) that sensory braille letter representations emerge in time before perceptual braille letter representations, analogous to the sequential processing of sensory representations before perceptual representations in the visual (*Isik et al., 2014*; *Carlson et al., 2013*; *Cichy et al., 2014*; *Carlson et al., 2011b*) and auditory (*Lowe et al., 2021*) domain.

To test H2, we conducted time-resolved within- and across-hand classification on EEG data. We determined the time point at which representations emerge by finding the first time point with respect to the onset of the braille stimulation where the classification effects are significant (50 consecutive significant time point criteria, 95% confidence intervals reported in brackets).

The EEG classification analyses revealed significant and reliable results for both within- and across-hand classification of braille letters, as well as their difference (*Figure 3A*; *N* = 11, 1000 bootstraps, one-tailed Wilcoxon signed-rank test, p < 0.05, FDR corrected). We found that within-hand classification became significant at 62 ms (29–111 ms) (*Figure 3A*, blue curve). To isolate sensory representations, we subtracted the results of the across-hand classification from the within-hand classification. This difference became significant at 77 ms (45–138 ms) (*Figure 3A*, black curve). In contrast, the across-hand classification, indicating perceptual representations, became significant later at 184 ms (127–230 ms) (*Figure 3A*, green curve).

Importantly, the temporal dynamics of sensory and perceptual representations differed significantly. Compared to sensory representations, the significance onset of perceptual representations was delayed by 107 ms (21–167 ms) (*N* = 11, 1000 bootstraps, one-tailed bootstrap test against zero, p = 0.012). This results pattern was consistent when defining the analysis baseline empirically (see *Figure 3—figure supplement 1*).

To approximate the sources of the temporal signals, we complemented the EEG classification analysis with a searchlight classification analysis in EEG sensor space. In the time window of highest decodability (~200–300 ms), sensory braille letter information was decodable from widespread electrodes across the scalp. In contrast, perceptual braille letter information was decodable later and from overall fewer electrodes which were located over right frontal, central, left parietal, and left temporal areas (*Figure 3B*; *N* = 11, one-tailed Wilcoxon signed-rank test, p < 0.05, FDR corrected across electrodes and time points). These additional results reinforce that sensory braille letter information is represented in more widespread brain areas than perceptual braille letter information, corroborating our ROI classification results.

In sum, our EEG results characterized the temporal dynamics with which braille letter representations emerge as a temporal sequence of sensory before perceptual representations.

## Relating representations of braille letters to behavior

The ultimate goal of perception is to provide an organism with representations enabling adaptive behavior. The analyses across space and time described above identified potential candidates for such braille letter representations in the sense that the representations distinguished between braille letters. However, not all of these representations have to be used by the brain to guide behavior; some of these representations might be epiphenomenal and only available to the experimenter (*deWit et al., 2016*; *Reddy and Kanwisher, 2007*; *Williams et al., 2007*).

Therefore, we tested the hypothesis (H3) that the sensory and perceptual braille letter representations identified in space (H1) and time (H2) are in a suitable format to be behaviorally relevant. We used perceived similarity as a proxy for behavioral relevance. The idea is that if two stimuli are perceived to be similar, they will also elicit similar actions (*Cichy et al., 2019*; *Bankson et al., 2018*; *Mur et al., 2013*; *Charest et al., 2014*). For this, we acquired perceived similarity ratings from blind

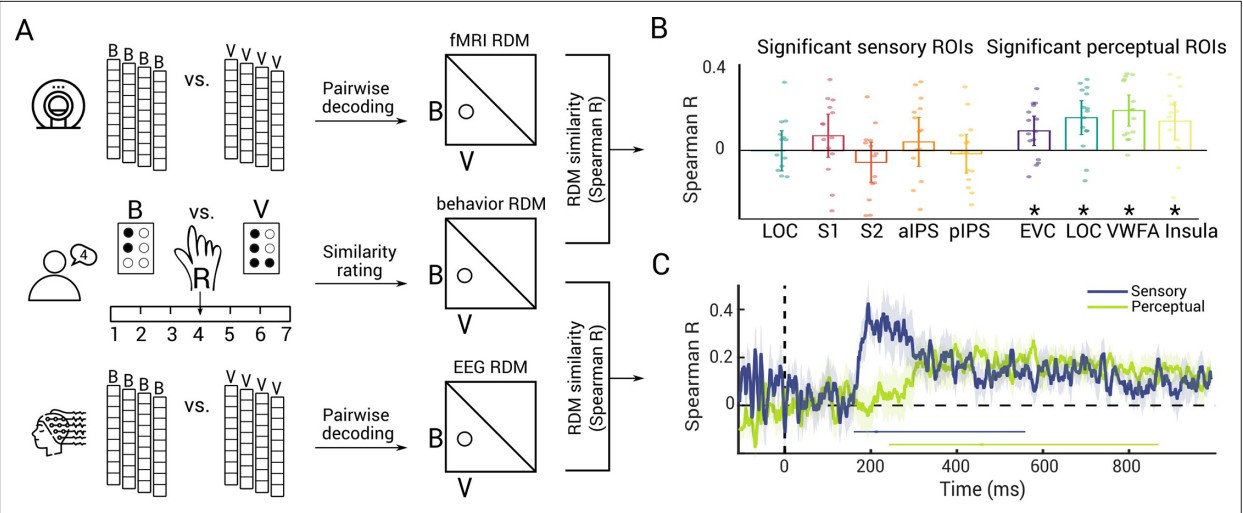

**Figure 4.** Representational similarity of braille letters in neural and behavioral measures. (**A**) For fMRI (top) and EEG (bottom), we formed representational dissimilarity matrices (RDMs) from pairwise decoding accuracies. In the behavioral experiment (middle), we presented two braille letters (e.g., B and V) on two braille cells. For all combinations, we asked participants to read both braille letters with the same hand (e.g., right) and verbally rate their similarity on a scale from 1 to 7. We formed the behavioral RDM from those perceptual similarity judgements for every letter pair. We then correlated the behavioral RDM (averaged over participants) with neural RDMs (subject specific) using Spearman's *R*. (**B**) Results of representational similarity analysis (RSA) relating fMRI and behavior in regions of interest (ROIs) that showed significant sensory (left) and perceptual (right) braille letter representations in fMRI (see *Figure 2*) (*N* = 15, two-tailed Wilcoxon signed-rank test, p < 0.05, false discovery rate [FDR] corrected). Stars below bars indicate significance above chance. Error bars represent 95% confidence intervals. Dots represent single subject data. (**C**) Results of RSA relating EEG and behavior for sensory (blue) and perceptual (green) representations. Shaded areas around curves indicate standard error of the mean. Significance onsets and their 95% confidence intervals are indicated by dots and lines below curves (*N* = 11, 1000 bootstraps, one-tailed Wilcoxon signed-rank test, p < 0.05, FDR corrected, color-coded as result curves).

The online version of this article includes the following figure supplement(s) for figure 4:

**Figure supplement 1.** Representational similarity of braille letters in neural and behavioral measures using Jaccard similarity.

**Figure supplement 2.** EEG-behavior representational similarity analysis (RSA) decoding results when only using occipital and parieto-occipital electrodes.

participants (*N* = 19) in a separate behavioral experiment (*Figure 4A*, middle) in which participants verbally rated the similarity of each pair of braille letters from the stimulus set on a scale.

To relate perceived similarity ratings to neural representations, we used representational similarity analysis (RSA) (*Kriegeskorte et al., 2008*). RSA relates different measurement spaces (such as EEG, fMRI, and behavior) by abstracting them into a common representational similarity space. We sorted behavioral similarity ratings into representational dissimilarity matrices (RDMs) indexed in rows and columns by the eight braille letters used in experimental conditions (*Figure 4A*). For neural data, we sorted decoding accuracies from the previous analyses as a dissimilarity measure (*Cichy et al., 2014*; *Guggenmos et al., 2018*). Thus, by arranging the classification results of the EEG and fMRI results we obtained EEG RDMs for every time point and fMRI RDMs for each ROI (for the within- and the across-hand analyses separately). To finally relate behavioral and neural data in the common RDM space, we correlated the behavioral RDM with each fMRI ROI RDM and each EEG time point RDM.

Considering braille letter representations in space through fMRI (*Figure 4B*, *N* = 15, one-tailed Wilcoxon signed-rank test, p < 0.05, FDR corrected), we found that previously identified perceptual representations (i.e., identified by the across-hand analysis) in EVC, LOC, VWFA, and insula showed significant correlations with behavior. In contrast, sensory representations (i.e., identified by the differences between within- and across-hand analyses) in S1, S2, aIPS, pIPS, and LOC were not significantly correlated with behavior. This indicates that perceptual braille letter representations in sighted reading areas are suitably formatted to guide behavior.

Considering braille letter representations in time through EEG (*Figure 4C*; *N* = 11, 1000 bootstraps, one-tailed Wilcoxon signed-rank test, p < 0.05, FDR corrected), we found significant relationships with behavior for both sensory and perceptual representations. The temporal dynamics mirrored those of the EEG classification analysis (*Figure 3A*), in that the results related to sensory representations

emerged earlier at 220 ms (167–567 ms) than the results related to perceptual representations at 466 ms (249–877 ms). The onset latency differences of 240 ms (−81 to 636 ms) ($N$ = 11, 1000 bootstrap, one-tailed bootstrap test against zero, p = 0.046) was significant. This indicates that both earlier sensory representations and later perceptual representations of braille letters are suitable formatted to guide behavior.

In sum, our RSA results highlighted that perceptual representations in sighted reading areas, as well as initial sensory and later perceptual representations in time, are suitably formatted to guide behavior.

## Discussion

We assessed experience-based brain plasticity at its boundaries by investigating the nature of braille information processing in the visually deprived brain. For this, we assessed the transformation of sensory to perceptual braille letter representations in blind participants. Our experimental strategy combining fMRI, EEG, and behavioral assessment yielded three key findings about spatial distribution, temporal emergence, and behavioral relevance. First, concerning the spatial distribution of braille letter representations, we found that sensory braille letter representations are located in tactile processing areas while perceptual braille letter representations are located in sighted reading areas. Second, concerning the temporal emergence of braille letter representations, we found that sensory braille letter representations emerge before perceptual braille letter representations. Third, concerning the behavioral relevance of representations, we found that perceptual representations identified in sighted reading areas, as well as sensory and perceptual representations identified in time, are suitably formatted to guide behavior.

### The topography of sensory and perceptual braille letter representations

Previous research has identified the regions activated during braille reading in high detail (*Büchel et al., 1998*; *Burton et al., 2012*; *Rączy et al., 2019*; *Reich et al., 2011*; *Sadato et al., 1998*; *Uhl et al., 1991*; *Sadato, 1996*). However, activation in a brain region alone does not indicate its functional role or the kind of information it represents (*Haynes and Rees, 2006*). Here, we characterize the information represented in a region by distinguishing between sensory and perceptual representations of single braille letters. Our findings extend our understanding of the cortical regions processing braille letters in the visually deprived brain in five ways.

First, we clarified the role of EVC activations in braille reading (*Büchel et al., 1998*; *Burton et al., 2012*; *Rączy et al., 2019*; *Reich et al., 2011*; *Sadato et al., 1998*; *Uhl et al., 1991*; *Sadato, 1996*) by showing that EVC harbors representations of single braille letters. More specifically, our finding that EVC represents perceptual rather than sensory braille letter information indicates that EVC representations are formatted at a higher perceptual level rather than a tactile input level. Previous studies also found that EVC of blind participants processes other higher-level information such as natural sounds (*Vetter et al., 2020*; *Mattioni et al., 2020*) and language (*Burton et al., 2002*; *Burton et al., 2003b*; *Röder et al., 2002*; *Bedny et al., 2011*; *Lane et al., 2015*; *Dietrich et al., 2013*; *Watkins et al., 2022*; *Abboud et al., 2019*; *Amedi et al., 2003*; *Burton, 2003a*). A parsimonious view is that EVC in the visually deprived brain engages in higher-level computations shared across domains, rather than performing multiple distinct lower-level sensory computations. Importantly, higher-level computations are not limited to the EVC in visually deprived brains. Natural sound representations (*Vetter et al., 2020*) and language activations (*Seydell-Greenwald et al., 2023*) are also located in EVC of sighted participants. This suggests that EVC, in general, has the capacity to process higher-level information (*Pascual-Leone and Hamilton, 2001*). Thus, EVC in the visually deprived brain might not be undergoing fundamental changes in brain organization (*Seydell-Greenwald et al., 2023*). This promotes a view of brain plasticity in which the cortex is capable of dynamic adjustments within pre-existing computational capacity limits (*Bedny, 2017*; *Seydell-Greenwald et al., 2023*; *Pascual-Leone and Hamilton, 2001*; *Makin and Krakauer, 2023*).

We note that our findings contribute additional evidence but cannot conclusively distinguish between the competing hypotheses that visually deprived brains dynamically adjust to the environmental constraints versus that they undergo a profound cortical reorganization. Resolving this debate

would require an analogous experiment in sighted people which taps into the same mechanisms as the present study. Defining a suitable control experiment is, however, difficult. Any other type of reading would likely tap into different mechanism than braille reading. Furthermore, whenever sighted participants are asked to perform a haptic reading task, outcomes can be confounded by visual imagery driving visual cortex (*Dijkstra et al., 2019*). Thus, the results would remain ambiguous as to whether observed differences between the groups index different mechanisms or plastic changes in the same mechanisms. Last, matching groups of sighted readers and braille readers such that they only differ with regard to their input modality seems practically unfeasible: There are vast differences within the blind population in general (for example, etiologies, onset, and severity), and the subsample of congenitally blind braille readers more specifically (for example, the quality and quantity of their braille education) as well as across braille and print readers (for example, different passive exposure to braille versus written letters during childhood *Englebretson et al., 2023*; *Merabet and Pascual-Leone, 2010*).

Second, we found that VWFA contains perceptual but not sensory braille letter representations. By clarifying the representational format of language representations in VWFA, our results support previous findings of the VWFA being functionally selective for letter and word stimuli in the visually deprived brain (*Reich et al., 2011*; *Striem-Amit et al., 2012*; *Liu et al., 2023*). Together, these findings suggest that the functional organization of the VWFA is modality-independent (*Reich et al., 2011*) depicting an important contribution to the ongoing debate on how visual experience shapes representations along the ventral stream (*Bedny, 2021*).

Third, LOC represented hand-dependent and -independent braille letter information, suggesting a 'hinge' function between sensory and perceptual braille letter representations. We stipulate that shape serves as an intermediate level representational format in between lower-level properties such as specific location of tactile stimulation or dot number in braille letters and higher-level perceptual letter features (*Amedi et al., 2001*).

Fourth, the finding of letter representations of tactile origin in both VWFA and LOC indicate that the functional organization of both regions is multimodal, contributing to the debate on how experience from vision or other sensory modalities shapes representations along the ventral stream (*Bedny, 2021*; *Amedi et al., 2001*).

Fifth, we observed that the somatosensory cortices and intra-parietal sulci represent hand-dependent but not hand-independent braille letter representations. This is consistent with previous studies reporting that the primary somatosensory cortex represents the location of tactile stimulation (*Sanchez-Panchuelo et al., 2012*), but not the identity of braille words (*Liu et al., 2023*). Taken together, these findings suggest that these tactile processing areas represent sensory rather than higher-level features of tactile inputs in visually deprived brains.

The involvement of the insula in processing braille letter information is more difficult to interpret. Based on previous studies in the sighted brain, the insula plays a role in tactile memory (*Bonda et al., 1996*; *Burton and Sinclair, 2000*; *Reed et al., 2004*) and multisensory integration (*Banati et al., 2000*; *Hadjikhani and Roland, 1998*; *Prather et al., 2004*; *Naghavi et al., 2007*). Both aspects could have contributed to our findings as braille letters are retrievable from long-term memory but are also inherently nameable and linked to auditory experiences. A future study could disambiguate the contributions of tactile memory and multisensory integration by presenting meaningless dot arrays, that are either unnamable or paired with invented names. Insular representations of trained, unnamable stimuli but not novel, unnamable stimuli would align with memory requirements. Insular representations of trained, namable stimuli but not trained, unnamable stimuli would favor audio-tactile integration.

Overall, identifying braille letter representations in widespread brain areas raises the question of how information flow is organized in the visually deprived brain. Functional connectivity studies report deprivation-driven changes of thalamo-cortical connections which could explain both arrival of information to and further flow of information beyond EVC. First, the coexistence of early thalamic connections to both S1 and V1 (*Müller et al., 2019*) would enable EVC to receive from different sources and at different time points. Second, potentially overlapping connections from both sensory cortices to other visual or parietal areas (*Ioannides et al., 2013*) could enable the visually deprived brain to process information in a widespread and interconnected array of brain areas. In such a network architecture, several brain areas receive and forward information at the same time. In contrast to

information discretely traveling from one processing unit to the next in the sighted brain's processing cascade, we can rather picture information flowing in a spatially and functionally more distributed and overlapping fashion.

## Sensory representations emerge before perceptual representations

Using time-resolved multivariate analysis of EEG data (*Isik et al., 2014*; *Carlson et al., 2013*; *Carlson et al., 2011b*), we showed that hand-dependent, sensory braille letter representations emerge in time before hand-independent, perceptual representations. Such sequential multi-step processing in time is a general principle of information processing in the human brain, also known in the visual (*Isik et al., 2014*; *Carlson et al., 2013*; *Cichy et al., 2014*) and auditory (*Lowe et al., 2021*) domain. Together, these findings suggest that the human brain, even in extreme instances of species-atypical cortical plasticity, honors this principle.

While braille letter reading follows the same temporal processing sequence as its visual counter-part, it operates on a different time scale. Our results indicate that braille letter classification peaks substantially later in time (~200 ms for hand-dependent and ~390 ms for hand-independent represen-tations) than previously reported classification of visually presented words, letters, objects, and object categories (e.g., ~125 ms for location-dependent and ~175 ms for location-independent representa-tions) (*Isik et al., 2014*; *Carlson et al., 2013*; *Cichy et al., 2014*; *Ling et al., 2019*; *Teng et al., 2015*). This discrepancy raises the question which factors could limit the speed of processing braille letters. Importantly, this delay is not a consequence of slower cortical processing in the tactile domain (*Müller et al., 2019*). We find that tactile information reaches the cortex fast: we can classify which hand was stimulated as early as 35 ms after stimulation onset (*Figure 3—figure supplement 2*). Thus, the delay relates directly to the identification of braille letters.

A compelling explanation for the temporal processing properties of braille letter information are the underlying reading mechanics. Braille reading is slower than print reading (*Bola et al., 2016*; *Brys-baert, 2019*) even if participants are fluent braille readers. This slowing is specific to braille reading and does not translate to other types of information intake in the visually deprived brain, for example, auditory information. Blind participants have higher listening rates (*Bragg et al., 2018*) and better auditory discrimination skills (*Hötting and Röder, 2009*) than sighted participants, indicating more efficient auditory processing (*Röder et al., 1996*; *Muchnik et al., 1991*). Together, this result pattern suggests that the temporal dynamics with which braille letter representations emerge are limited by the particular efficiency of the braille letter system, rather than the capacity of the brain. To test this idea, future studies could compare the temporal dynamics of braille letter and haptic object represen-tations: a temporal processing delay specific to braille letters would support the hypothesis.

## Representations identified in space and time guide behavior

Our results clarified that perceptual rather than sensory braille letter representations identified in space are suitably formatted to guide behavior. However, we only use one specific task to assess behavior and, therefore, acknowledge that this finding is task-dependent. Arguably, general similarity ratings of braille letters depend more on intake-independent (e.g., dot arrangements or linguistic similarities such as pronunciation) than intake-dependent features (similarities in stimulation location on finger). Future behavioral assessments could ask participants to assess similarity separately based on only stimulation location or linguistic features. We would predict that similarity ratings based on stimulation location are related to sensory representations while similarity ratings based on linguistic features are related to perceptual representations of braille letters.

Concerning temporal dynamics, our results reveal that sensory representations of braille letters are relevant for behavior earlier than perceptual ones. Interestingly, the similarity between perceptual braille letter representations and behavioral similarity ratings emerges in a time window in which transcranial magnetic stimulation (TMS) over the VWFA affects braille letter reading in sighted braille readers (*Bola et al., 2019*). This implies that around 320–420 ms after the onset of reading braille, visually deprived and sighted brains utilize braille letter representations for performing tasks such as letter identification. Applying a comparable TMS protocol not only to sighted but also non-sighted braille readers would elucidate whether this time window of behavioral relevance can be generalized to braille reading, independent of visual experience.

Similarity ratings and sensory representations as captured by EEG are correlated, and so are similarity ratings and representations in perceptual ROIs, but not sensory ROIs. This might be interpreted as suggesting a link between the sensory representations captured in EEG and the representations in perceptual ROIs. However, we do not have any evidence towards this idea. Differing signal-to-noise ratios (SNRs) for the different ROIs and sensory versus perceptual analysis could be an alternative explanation.

## Conclusions

Our investigation of experience-based plasticity at its boundaries due to the loss of the visual modality reveals a nuanced picture of its potential and limits. On the one hand, our findings emphasize how plastic the brain is by showing that regions typically processing visual information adapt to represent perceptual braille letter information. On the other hand, our findings illustrate inherent limits of brain plasticity. Brain areas represent information from atypical inputs within the boundaries of their pre-existing computational capacity and the progression from sensory to perceptual transformations adheres to a common temporal scheme and functional role.

## Methods

### Participants

We conducted three separate experiments with partially overlapping participants: an fMRI, an EEG, and a behavioral experiment. All experiments were approved by the ethics committee of the Department of Education and Psychology of the Freie Universität Berlin and were conducted in accordance with the Declaration of Helsinki. Sixteen participants completed the fMRI experiment. One person was excluded due to technical problems during the recording, leaving a total of 15 participants in the fMRI experiment (mean age 39 years, SD = 10, 9 females). Eleven participants participated in the EEG experiment ($N$ = 11, mean age 44 years, SD = 10, 8 females). The participant pools of the EEG and fMRI experiments overlapped by five participants. Out of a total of 21 participants, 19 participants (excluding one fMRI and one EEG participant) completed an additional behavioral task in which they rated the perceived similarity of braille letter pairs. All participants were blind since birth or early childhood (≤3 years, for details see *Supplementary file 1*). All participants provided informed consent prior to the studies and received a monetary reward for their participation.

### Experimental stimuli and design

In all experiments, we presented braille letters (B, C, D, L, M, N, V, and Z; *Figure 2A*) to the left and right index fingers of participants using piezo-actuated refreshable braille cells (https://metec-ag.de/index.php) with two modules of eight pins each. We only used the top 6 pins from each module to present letters from the braille alphabet. The modules were taped to the clothes of a participant for the fMRI experiment and on the table for the EEG and behavioral experiment. This way, participants could read in a comfortable position with their index fingers resting on the braille cells to avoid motion confounds. Importantly, our experimental setup did not involve tactile exploration or sliding motions. We instructed participants to read letters regardless of whether the pins passively stimulated their right or left index finger. We presented all eight letters to both hands, resulting in 16 experimental conditions (8 letters × 2 hands). In addition, two braille letters (E and O) were included as catch stimuli and participants were instructed to respond to them by pressing a button (fMRI) or pedal (EEG) with their foot. Catch trials were excluded from further analysis due to confounding motor and sensory signals.

### Experimental procedures

#### fMRI experiment

The fMRI experiment consisted of two sessions. Fifteen participants completed the first fMRI session, during which we recorded a structural image (~4 min), a localizer run (7 min), and 10 runs of the main experiment (56 min). The total duration of the first session was 67 min excluding breaks. Eight of these 15 subjects completed a second fMRI recording session, in which we recorded an additional 15 runs of the main experiment (85 min). We did not record any structural images or localizer runs in the second session, resulting in a total duration of 85 min excluding breaks.

### fMRI main experiment

During the fMRI main experiment, we presented participants with letters on braille cells and asked them to respond to occasionally appearing catch letters. We presented letters for 500 ms, with a 2500-ms ISI (see *Figure 2B*, top). Each regular trial – belonging to one of the 16 experimental conditions – was repeated 5 times per run (run duration: 337 s) in random order. Regular trials were interspersed every ~20 trials with a catch trial, such that a catch trial occurred about once per minute. In addition, every third to fifth trial (equally probable) was a null trial where no stimulation was given. In total, one run consisted of 80 regular trials, 5 catch trials, and 22 null trials, amounting to a total of 107 trials per run.

To ensure that participants were able to read letters with both hands and understood the task instructions, participants first completed an experimental run outside the scanner.

### fMRI localizer experiment

To define ROIs, we performed a separate localizer experiment prior to the main fMRI experiment with tactile stimuli in four experimental conditions: braille letters read with the left hand, braille letters read with the right hand, fake letters read with the left hand, and fake letters read with the right hand. The letters presented in the braille conditions were 16 letters from the alphabet excluding the letters used in the main experiment. The stimuli in the fake letter conditions were 16 tactile stimuli that were each composed of 8 dots, deviating from the standard 6-dot configuration in the braille alphabet.

The localizer experiment consisted of a single run lasting 432 s, comprising five blocks of presentation of braille letters left, braille letters right, fake letters left, fake letters right, and blank blocks as baseline. Each stimulation block was 14.4 s long, consisting of 18 different letters presentations (500 ms on, 300 ms off) including two one-back repetitions that participants were instructed to respond to by pressing a button with their foot. We presented stimulation blocks in random order and regularly interspersed them with blank blocks.

### EEG experiment

The EEG experiment consisted of two sessions. Eleven participants completed the first session and eight of these participants completed a second session. The total duration of each session was 59 min excluding breaks.

The experimental setup was similar to that for fMRI but adapted to the specifics of EEG. We presented braille letters for 500 ms with a 500-ms ISI on regular trials. In catch trials, the letters were presented for 500 ms with a 1100-ms ISI to avoid contamination of movement on subsequent trials (see *Figure 2B*, bottom). Each of the 16 experimental conditions was presented 170 times per session. Regular trials were interspersed every fifth to seventh trial (equally probable) with a catch trial. In total, one EEG recording session consisted of 2720 regular trials and 541 catch trials, amounting to a total of 3261 trials. Two participants completed additional trials due to technical problems leading to a total of 190 and 180 repetitions per stimulus, accordingly.

### Braille screening task

Prior to the experiment, participants completed a short screening task during which each letter of the alphabet was presented for 500 ms to each hand in random order. Participants were asked to verbally report the letter they had perceived to assess their reading capabilities with both hands using the same presentation time as in the experiment. The average performance for the left hand was 89% correct (SD = 10) and for the right hand it was 88% correct (SD = 13).

### Behavioral letter similarity ratings

In a separate behavioral experiment, participants judged the perceived similarity of the braille letters used in the neuroimaging experiments. For this task, participants sat at a desk and were presented with two braille cells next to each other. Each pair of letters was presented once, and participants compared them with the same finger. We instructed participants to freely compare the similarity of pairs of Braille letters without specifying which parameters they should use for the similarity assessment. The rating was without time constraints, meaning participants decided when they rated the

stimuli. Participants were asked to verbally rate the similarity of each pair of braille letters on a scale from 1 = very similar to 7 = very different and the experimenter noted down their responses.

## fMRI data acquisition, preprocessing, and preparation

### fMRI acquisition

We acquired MRI data on a 3 T Siemens Tim Trio scanner with a 12-channel head coil. We obtained structural images using a T1-weighted sequence (magnetization-prepared rapid gradient-echo, 1 mm³ voxel size). For the main experiment and the localizer run, we obtained functional images covering the entire brain using a T2*-weighted gradient-echo planar sequence (repetition time TR = 2 s, echo time TE = 30 ms, flip angle = 70°, 3 mm³ voxel size, 37 slices, field of view FOV = 192 mm, matrix size = 64 × 64, interleaved acquisition).

### fMRI preprocessing

For fMRI preprocessing, we used tools from FMRIB's Software Library (FSL, http://www.fmrib.ox.ac.uk/fsl). We excluded non-brain tissue from analysis using the Brain Extraction Tool (BET) (*Smith, 2002*) and motion corrected the data using MCFLIRT (*Jenkinson et al., 2002*). We did not apply high- or low-pass temporal filters and did not perform slice time correction. We spatially smoothed fMRI localizer data with an 8-mm FWHM Gaussian kernel. We registered functional images to the high-resolution structural scans and to the MNI standard template using FLIRT (*Jenkinson and Smith, 2001*). We carried out all further fMRI analyses in MATLAB R2021a (https://www.mathworks.com/).

### Univariate fMRI analysis

For all univariate fMRI analyses, we used SPM12 (http://www.fil.ion.ucl.ac.uk/spm). For the main experiment, we modeled the fMRI responses to the 16 experimental conditions for each run using a general linear model (GLM). The onsets and durations of each image presentation entered the GLM as regressors and were convolved with a hemodynamic response function (hrf). Six movement parameters (pitch, yaw, roll, *x*-, *y*-, *z*-translation) entered the GLM as nuisance regressors. For each of the 16 conditions we converted GLM parameter estimates into *t*-values by contrasting each parameter estimate against the implicit baseline. This resulted in 16-condition-specific *t*-value maps per run and participant.

For the localizer experiment, we modeled the fMRI response to the five experimental conditions entering block onsets and durations as regressors of interest and movement parameters as nuisance regressors before convolving with the hrf. From the resulting three parameter estimates we generated two contrasts. The first contrast served to localize activations in primary (S1) and secondary (S2) somatosensory cortex and was defined as letters and fake letters > baseline. The second contrast served to localize activations in EVC, V4, LOC, LFA, VWFA, aIPS, pIPS, and insula and was defined as letters > fake letters. In sum, this resulted in two *t*-value maps for the localizer run per participant.

### Definition of ROIs

To identify regions along the sighted reading and tactile processing pathway, we defined ROIs in a two-step procedure. We first constrained ROIs by anatomical masks using brain atlases, in each case combining regions across both hemispheres. We included five ROIs from the sighted reading pathway: EVC (merging the anatomical masks of V1, V2, and V3), V4, LOC, LFA, and the VWFA. We also included five ROIs from the tactile processing pathway: S1, S2, aIPS (merging the anatomical masks of IPS3, IPS4, and IPS5), pIPS (merging the anatomical masks of IPS0, IPS1, and IPS2), and the insula. For EVC, V4, LOC, aIPS, and pIPS, we used masks from the probabilistic Wang atlas (*Wang et al., 2015*). For LFA, we defined the mask using the MarsBaR Toolbox (https://marsbar-toolbox.github.io/) with a 10-mm radius around the center voxel at MNI coordinates *X* = −40, *Y* = −78, and *Z* = −18 (*Thesen et al., 2012*). We also defined the VWFA mask using the MarsBaR Toolbox with a 10-mm radius around the center voxel at MNI coordinates *X* = −44, *Y* = −57, and *Z* = −13 (*Cohen et al., 2000*) and converted from Talairach to MNI space using the MNI<->Talairach Tool (https://bioimagesuiteweb.github.io/bisweb-manual/tools/mni2tal.html). We created the mask for S1 by merging the sub-masks of BA1, BA2, and BA3 from the WFU PickAtlas (https://www.nitrc.org/projects/wfu_pickatlas/) and the mask for S2 by merging the sub-masks operculum 1–4 from the Anatomy Toolbox (*Eickhoff et al.,*

*2005*). Lastly, we extracted the mask for the insula from the WFU PickAtlas. The smallest mask was V4 which included 321 voxels. Therefore, in a second step, we selected the 321 most activated voxels of the participant-specific localizer results within each of the masks, using the letters and fake letters > baseline contrast for S1 and S2 and the letters > fake letters for the remaining ROIs. This yielded participant-specific definitions for all ROIs.

## EEG data acquisition and preprocessing

We recorded EEG data using an EASYCAP 64-channel system and a Brainvision actiCHamp amplifier at a sampling rate of 1000 Hz. The electrodes were placed according to the standard 10-10 system. The data were filtered online between 0.03 and 100 Hz and re-referenced online to FCz.

We preprocessed data offline using the EEGLAB toolbox version 14 (*Delorme and Makeig, 2004*). We incorporated a low-pass filter with a cut-off at 50 Hz and epoched trials between −100 and 999 ms with respect to stimulus onset, resulting in 1100 1-ms data points per epoch. We baseline corrected the epochs by subtracting the mean of the 100-ms prestimulus time window from the epoch. We re-referenced the data offline to the average reference. To clean the data from artifacts such as eye blinks, eye movements, and muscular contractions, we used independent component analysis as implemented in the EEGLAB toolbox. We used SASICA (*Chaumon et al., 2015*) to guide the visual inspection of components for removal. We identified components related to horizontal eye movements using two lateral frontal electrodes (F7–F8). During five recordings (one participant first session, four participants second session), additional external electrodes were available that allowed for the direct recording of the horizontal electro-oculogram to identify and remove components related to horizontal eye movements. For blink artifact detection based on the vertical electro-oculogram, we used two frontal electrodes (Fp1 and Fp2). As a final step, we applied multivariate noise normalization to improve the SNR and reliability of the data (*Guggenmos et al., 2018*), resulting in subject-specific trial-based time courses of electrode activity.

## Braille letter classification from brain measurements

To determine the amount of information about braille letter identity present in brain measurements, we used a multivariate classification scheme (*Cichy et al., 2013*; *Cichy et al., 2011*; *Carlson et al., 2011a*; *Isik et al., 2014*). We conducted subject-specific braille letter classification in two ways. First, we classified between letter pairs presented to one reading hand, that is, we trained and tested a classifier on brain data recorded during the presentation of braille stimuli to the same hand (either the right or the left hand). This yields a measure of hand-dependent braille letter information in neural measurements. We refer to this analysis as within-hand classification. Second, we classified between letter pairs presented to different hands in that we trained a classifier on brain data recorded during the presentation of stimuli to one hand (e.g., right), and tested it on data related to the other hand (e.g., left). This yields a measure of hand-independent braille letter information in neural measurements. We refer to this analysis as across-hand classification. We tested both within- and across-hand pairwise classification accuracies against a chance level of 50%. We also calculated a within- to across-hand classification score which we compared against 0.

All classification analyses were carried out in MATLAB R2021a (https://www.mathworks.com/) and relied on binary c-support vector classification (C-SVC) with a linear kernel as implemented in the libsvm toolbox (*Chang and Lin, 2011*; https://www.csie.ntu.edu.tw/cjlin/libsvm). Furthermore, all analyses were conducted in a participant specific manner. The next section describes the multivariate fMRI and EEG analyses in more detail.

### Spatially resolved multivariate fMRI analysis

We conducted both an ROI-based and a spatially unbiased volumetric searchlight procedure (*Kriegeskorte et al., 2006*). For each ROI included in the ROI-based analysis, we extracted and arranged *t*-values into pattern vectors for each of the 16 conditions and experimental runs. If participants completed only one session, the analysis was conducted on 10 runs. If participants completed both sessions, the 10 runs from session 1 and 15 runs from session 2 were pooled and the analysis was conducted across 25 runs. To increase the SNR, we randomly assigned run-wise pattern vectors into bins and averaged them into pseudo-runs. For participants with one session, the bin size was two runs, resulting in five pseudo-runs. If participants completed 2 sessions and thus had 25 runs, the

bin size was 5 runs resulting in 5 pseudo-runs. Thus, in both cases, each participant ended up with five pseudo-run pattern vectors that entered the classification analysis. We then performed fivefold leave-one-pseudo-run-out-cross validation, training on four and testing on one pseudo-trial per classification iteration.

We will first describe the classification procedure for braille letters within-hand and then for the classification of braille letters across-hand.

## fMRI ROI-based classification of braille letters within-hand

For the classification of braille letters within-hand, we assigned four pseudo-trials corresponding to the data from two braille letters of the same hand (e.g., right) to the training set. We then tested the support vector machine (SVM) on the remaining, fifth pseudo-trial corresponding to data from the same two braille letters of the same hand (e.g., right) as in the training set but using held-out data for the testing set. This yielded percent classification accuracy (50% chance level) as output. Equivalent SVM training and testing were repeated for all combinations of letter pairs within each hand.

With 8 letters that were all classified pairwise once per hand, this resulted in 28 pairwise classification accuracies per hand. We averaged accuracies across condition pairs and hands, yielding a measure of hand-dependent braille letter information for each ROI and participant separately.

## fMRI ROI-based classification of braille letters across-hand

The classification procedure of braille letters across reading hands was identical to the classification procedure within-hand with the important difference that the training data always came from one hand (e.g., right) and the testing data from the other hand (e.g., left).

With 8 letters that were all classified pairwise once across two hands, this resulted again in 28 pairwise classification accuracies across-hand per training–testing direction (i.e., train left, test right, and vice versa). We averaged accuracies across condition pairs and training–testing directions, yielding a measure of hand-independent braille letter information for each ROI and participant separately.

## fMRI searchlight classification of braille letters

The searchlight procedure was conceptually equivalent to the ROI-based analysis. For each voxel $v_i$ in the 3D $t$-value maps, we defined a sphere with a radius of four voxels centered around voxel $v_i$. For each condition and run, we extracted and arranged the $t$-values for each voxel of the sphere into pattern vectors. Classification of braille letters across-hand proceeded as described above. This resulted in one average classification accuracy for voxel $v_i$. Iterated across all voxels this yielded a 3D volume of classification accuracies across the brain for each participant separately.

## Time-resolved classification of braille letters within-hand from EEG data

To determine the timing with which braille letter information emerges in the brain, we conducted time-resolved EEG classification (*Isik et al., 2014*; *Carlson et al., 2011b*). This procedure was conceptually equivalent to the fMRI braille letter classification in that it classified letter pairs either within- or across-hand and was conducted separately for each participant.

For each time point of the epoched EEG data, we extracted 63 EEG channel activations and arranged them into pattern vectors for each of the 16 conditions. Participants who completed one session had 170 trials per condition and participants who completed two sessions had 340 trials per condition. To increase the SNR, we randomly assigned the trials into bins and averaged them into new pseudo-trials. For participants with one session, the bin size was 34 trials, resulting in 5 pseudo-trials. If participants completed 2 sessions and thus had 340 trials, the bin size was 68 trials resulting in 5 pseudo-trials. In both cases, each participant ended up with five pseudo-run pattern vectors that entered the classification analysis. We then performed fivefold leave-one-pseudo-run-out-cross validation, training on four and testing on one pseudo-trial per classification iteration. This procedure was repeated 100 times with random assignment of trials to pseudo-trials, and across all combinations of letter pairs and hands. We averaged results across condition pairs, folds, iterations, and hands, yielding a decoding accuracy time course reflecting how much hand-dependent braille letter information was present at each time point in each participant.

### Time-resolved classification of braille letters across-hand from EEG data

The classification procedure for braille letters across-hand was identical to the classification of braille letters within-hand with the crucial difference that training and testing data always came from separate hands and results were averaged across condition pairs, folds, iterations, and training–testing directions. Averaging results yielded a decoding accuracy time course reflecting how much hand-independent braille letter information was present at each time point in each participant.

### Time-resolved EEG searchlight in sensor space

We conducted an EEG searchlight analysis resolved in time and sensor space (i.e., across 63 EEG channels) to gain insights into which EEG channels contributed to the results of the time-resolved analysis described above. For the EEG searchlight, we conducted the time-resolved EEG classification as described above with the following difference: For each EEG channel $c_i$, we conducted the classification procedure on the four closest channels surrounding c. The classification accuracy was stored at the position of c. After iterating across all channels and down-sampling the time points to a 10-ms resolution, this yielded a classification accuracy map across all channels and time points in 10 ms steps for each participant.

## RSA of brain data and behavioral letter similarity ratings

To determine the subset of neural braille letter representations identified that is relevant for behavior (*Cichy et al., 2019*; *Bankson et al., 2018*; *Mur et al., 2013*; *Charest et al., 2014*), we compared perceptual letter similarity ratings to braille letter representations identified from EEG and fMRI signals using RSA (*Kriegeskorte et al., 2008*). RSA characterizes the representational space of a measurement space (e.g., fMRI or EEG data) with an RDM. RDMs aggregate pairwise distances between responses to all experimental conditions, thereby abstracting from the activity patterns of measurement units (e.g., fMRI voxels or EEG channels) to between-condition dissimilarities. The rationale of the approach is that neural measures and behavior are linked if their RDMs are similar.

We constructed RDMs for behavior, fMRI and EEG as follows.

For behavior, we arranged the perceptual similarity judgments averaged across participants (indicated by participants on a scale from 1 = very similar to 7 = very different) into an RDM format. All RDMs were averaged over both hands and had the dimensions 8 letters × 8 letters.

For both fMRI and EEG, we used the classification results from the conducted within- and across-hand classifications as a measure of (dis-)similarity relations between braille letters. Classification accuracies can be interpreted as a measure of dissimilarity because two conditions have a higher classification accuracy when they are more dissimilar (*Cichy et al., 2014*; *Guggenmos et al., 2018*). Thus, we assembled participant-specific RDMs for each fMRI ROI and EEG time point in the time course from decoding accuracies.

In a final step we correlated (Spearman's *R*) the lower triangular part of the respective RDMs (without the diagonal; *Jenkinson and Smith, 2001*). For fMRI, this resulted in one correlation value per participant and ROI. For EEG, this analysis resulted in one correlation time course per participant.

## Statistical testing

### Wilcoxon signed-rank test

We performed non-parametric one-tailed Wilcoxon signed-rank tests to test for above-chance classification accuracy for ROIs in the fMRI classification, for time points in the EEG classification, for time points and channels in the EEG searchlight, and for ROI and time courses in the RSA. In each case, the null hypothesis was that the observed parameter (i.e., classification accuracy, correlation) came from a distribution with a median of chance level performance (i.e., 50% for pairwise classification; 0 correlation). The resulting p-values were corrected for multiple comparisons using the FDR (*Benjamini and Hochberg, 1995*) at 5% level if more than one test was conducted. This was done (1) across ROIs in the ROI classification and fMRI-behavior RSA, (2) across time points in the EEG classification and EEG-behavior RSA, and (3) across time points and channels in the EEG searchlight.

## Bootstrap tests

We used bootstrapping to compute 95% confidence intervals for onset latencies (the first 50 consecutive significant time points after trial onset) of EEG time courses as well as for determining the significance of onset latencies. In each case, we sampled the participant pool 1000 times with replacement calculated the statistic of interest for each sample.

For the EEG onset latency differences, we bootstrapped the latency difference between the onsets of the time courses of hand-dependent or -independent letter representations. This yielded an empirical distribution that could be compared to zero. To determine whether onset latencies differences in the EEG time courses were significantly different from zero, we computed the proportion of values that were equal to or smaller than zero and corrected them for multiple comparisons using FDR at p = 0.05.

## Other statistical tests

For the fMRI searchlight classification results, we applied a voxel-wise height threshold of p = 0.001. The resulting p-values were corrected for multiple comparisons using the FWE at the 5% level.

## Code availability

The code used in this study is available on Github via https://github.com/marleenhaupt/BrailleLetterRepresentations/ (copy archived at *Haupt, 2024*).

## Acknowledgements

We thank all of our participants for taking part in our experiments. We also thank Agnessa Karapetian, Johannes Singer, and Siying Xie for their valuable comments on the manuscript. We acquired EEG and fMRI data at the Center for Cognitive Neuroscience (CCNB), Freie Universität Berlin, and we thank the HPC Service of ZEDAT, Freie Universität Berlin, for computing time (*Bennett et al., 2020*).The study was supported by German Research Council grants (CI241/1-1, CI241/3-1, CI241/7-1) and European Research Council grants (ERC-StG-2018-803370) to R.M.C. The funders had no role in study design, data collection, and analysis, decision to publish or preparation of the manuscript.

## Additional information

### Funding

| Funder | Grant reference number | Author |
|---|---|---|
| Deutsche Forschungsgemeinschaft | CI241/1-1 | Radoslaw Cichy |
| Deutsche Forschungsgemeinschaft | CI241/3-1 | Radoslaw Cichy |
| Deutsche Forschungsgemeinschaft | CI241/7-1 | Radoslaw Cichy |
| Deutsche Forschungsgemeinschaft | INST 272/297-1 | Radoslaw Cichy |
| European Research Council | ERC-StG-2018-803370 | Radoslaw Cichy |

The funders had no role in study design, data collection, and interpretation, or the decision to submit the work for publication.

### Author contributions

Marleen Haupt, Software, Formal analysis, Visualization, Methodology, Writing – original draft, Writing – review and editing; Monika Graumann, Conceptualization, Data curation, Software, Formal analysis, Investigation, Methodology, Project administration; Santani Teng, Conceptualization, Supervision, Writing – review and editing; Carina Kaltenbach, Data curation, Investigation; Radoslaw Cichy,

Conceptualization, Resources, Supervision, Funding acquisition, Project administration, Writing – review and editing

**Author ORCIDs**
Marleen Haupt ⓘ https://orcid.org/0000-0003-1683-8679

**Ethics**
All experiments were approved by the ethics committee of the Department of Education and Psychology of the Freie Universität Berlin and were conducted in accordance with the Declaration of Helsinki. All participants provided informed consent prior to the studies and received a monetary reward for their participation.

Reviewer #1 (Public review): https://doi.org/10.7554/eLife.98148.3.sa1
Reviewer #2 (Public review): https://doi.org/10.7554/eLife.98148.3.sa2
Author response https://doi.org/10.7554/eLife.98148.3.sa3

## Additional files

**Supplementary files**
• Supplementary file 1. Participant information.
• MDAR checklist

**Data availability**

The raw fMRI and EEG data are available on OpenNeuro via https://openneuro.org/datasets/ds004956 and https://openneuro.org/datasets/ds004951/. The preprocessed fMRI, EEG, and behavioral data as well as the results of the ROI classification, time classification, and RSA can be accessed on OSF via https://osf.io/a64hp/.

The following datasets were generated:

| Author(s) | Year | Dataset title | Dataset URL | Database and Identifier |
|---|---|---|---|---|
| Haupt M, Graumann M, Teng S, Kaltenbach C, MCichy R | 2024 | Braille letters - fMRI | https://doi.org/10.18112/openneuro.ds004956.v1.0.1 | OpenNeuro, 10.18112/openneuro.ds004956.v1.0.1 |
| Haupt M, Graumann M, Teng S, MCichy R | 2024 | Braille letters - EEG | https://doi.org/10.18112/openneuro.ds004951.v1.0.0 | OpenNeuro, 10.18112/openneuro.ds004951.v1.0.0 |
| Haupt M | 2024 | The transformation of sensory to perceptual braille letter representations in the visually deprived brain | https://doi.org/10.17605/OSF.IO/A64HP | Open Science Framework, 10.17605/OSF.IO/A64HP |

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
