## [Editor Report · eLife Assessment]

This **valuable** study investigates the brain representations of Braille letters in blind participants and provides evidence using EEG and fMRI that the decoding of letter identity across the reading hand takes place in the visual cortex. The evidence supporting the claims of the authors is **convincing** and the work will be of interest to neuroscientists working on brain plasticity.

---

## [Referee Report · Reviewer #1 (Public review)]

Summary:

The researchers examined how individuals who were born blind or lost their vision early in life process information, specifically focusing on the decoding of Braille characters. They explored the transition of Braille character information from tactile sensory inputs, based on which hand was used for reading, to perceptual representations that are not dependent on the reading hand.

They identified tactile sensory representations in areas responsible for touch processing and perceptual representations in brain regions typically involved in visual reading, with the lateral occipital complex serving as a pivotal "hinge" region between them.

In terms of temporal information processing, they discovered that tactile sensory representations occur prior to cognitive perceptual representations. The researchers suggest that this pattern indicates that even in situations of significant brain adaptability, there is a consistent chronological progression from sensory to cognitive processing.

Strengths:

By combining fMRI and EEG, and focusing on the diagnostic case of Braille reading, the paper provides an integrated view of the transformation processing from sensation to perception in the visually deprived brain. Such a multimodal approach is still rare in the study of human brain plasticity and allows to discern the nature of information processing in blind people early visual cortex, as well as the timecourse of information processing in a situation of significant brain adaptability.

Weaknesses:

ROI and searchlight analyses are not completely overlapping, although this might be due to the specific limits and strengths of each approach. Moreover, the conclusions regarding the behavioral relevance of the sensory and perceptual representations in the putatively reorganized brain, although important, are limited due to the behavioral measurements adopted.

---

## [Referee Report · Reviewer #2 (Public review)]

Summary:

Haupt and colleagues performed a well-designed study to test the spatial and temporal gradient of perceiving braille letters in blind individuals. Using cross-hand decoding of the read letters, and comparing it to the decoding of the read letter for each hand, they defined perceptual and sensory responses. Then they compared where (using fMRI) and when (using EEG) these were decodable. Using fMRI, they showed that low-level tactile responses specific to each hand are decodable from the primary and secondary somatosensory cortex as well as from IPS subregions, the insula and LOC. In contrast, more abstract representations of the braille letter independent from the reading hand were decodable from several visual ROIs, LOC, VWFA and surprisingly also EVC. Using a parallel EEG design, they showed that sensory hand-specific responses emerge in time before perceptual braille letter representations. Last, they used RSA to show that the behavioral similarity of the letter pairs correlates to the neural signal of both fMRI (for the perceptual decoding, in visual and ventral ROIs) and EEG (for both sensory and perceptual decoding).

Strengths:

This is a very well-designed study and it is analyzed well. The writing clearly describes the analyses and results. Overall, the study provides convincing evidence from EEG and fMRI that the decoding of letter identity across the reading hand occurs in the visual cortex in blindness. Further, it addresses important questions about the visual cortex hierarchy in blindness (whether it parallels that of the sighted brain or is inverted) and its link to braille reading.

---

## [Author Response]

The following is the authors’ response to the original reviews.

**Public Reviews:**

**Reviewer #1 (Public Review):**
We thank Reviewer #1 for the relevant and insightful comments on our paper. Please find our detailed answers below in the Recommendations to the Authors section.Summary:The researchers examined how individuals who were born blind or lost their vision early in life process information, specifically focusing on the decoding of Braille characters. They explored the transition of Braille character information from tactile sensory inputs, based on which hand was used for reading, to perceptual representations that are not dependent on the reading hand.They identified tactile sensory representations in areas responsible for touch processing and perceptual representations in brain regions typically involved in visual reading, with the lateral occipital complex serving as a pivotal "hinge" region between them.In terms of temporal information processing, they discovered that tactile sensory representations occur prior to cognitive-perceptual representations. The researchers suggest that this pattern indicates that even in situations of significant brain adaptability, there is a consistent chronological progression from sensory to cognitive processing.Strengths:By combining fMRI and EEG, and focusing on the diagnostic case of Braille reading, the paper provides an integrated view of the transformation processing from sensation to perception in the visually deprived brain. Such a multimodal approach is still rare in the study of human brain plasticity and allows us to discern the nature of information processing in blind people's early visual cortex, as well as the time course of information processing in a situation of significant brain adaptability.Weaknesses:The lack of a sighted control group limits the interpretations of the results in terms of profound cortical reorganization, or simple unmasking of the architectural potentials already present in the normally developing brain.

We thank the reviewer for raising this important point! We acknowledge that our claims regarding the unmasking of architectural potentials in both the normally developing and visually deprived brain are limited by the study design we employed. However, we note that defining an appropriate control group and assessing non-visual reading in sighted participants is far from straightforward. We discuss these issues in our response to the Public Review of Reviewer 2.

Moreover, the conclusions regarding the behavioral relevance of the sensory and perceptual representations in the putatively reorganized brain are limited due to the behavioral measurements adopted.

We agree with the reviewer that the relation between behavior and neural representations as established via perceived similarity judgments are task-dependent, and that a richer assessment of behavior would be valuable. Please note, however, that this limitation pertains to any experimental task used to assess behavior in the laboratory. Our major goal was to assess whether the identified neural representations are suitably formatted to be used by the brain for at least one behavior rather than being epiphenomenal. We found that the representations are suitably formatted for similarity judgments, thus establishing that they are relevant for at least this behavior. We also argue that judging similarity is a complex task that may underlie many other relevant behaviors. We discuss this point further in response to the Recommendations to the Authors.

**Reviewer #2 (Public Review):**

We thank the reviewer for the considerate and thoughtful suggestions. Please find a detailed description of the implemented changes below.

Summary:Haupt and colleagues performed a well-designed study to test the spatial and temporal gradient of perceiving braille letters in blind individuals. Using cross-hand decoding of the read letters, and comparing it to the decoding of the read letter for each hand, they defined perceptual and sensory responses. Then they compared where (using fMRI) and when (using EEG) these were decodable. Using fMRI, they showed that low-level tactile responses specific to each hand are decodable from the primary and secondary somatosensory cortex as well as from IPS subregions, the insula, and LOC. In contrast, more abstract representations of the braille letter independent from the reading hand were decodable from several visual ROIs, LOC, VWFA, and surprisingly also EVC. Using a parallel EEG design, they showed that sensory hand-specific responses emerge in time before perceptual braille letter representations. Last, they used RSA to show that the behavioral similarity of the letter pairs correlates to the neural signal of both fMRI (for the perceptual decoding, in visual and ventral ROIs) and EEG (for both sensory and perceptual decoding).Strengths:This is a very well-designed study and it is analyzed well. The writing clearly describes the analyses and results. Overall, the study provides convincing evidence from EEG and fMRI that the decoding of letter identity across the reading hand occurs in the visual cortex in blindness. Further, it addresses important questions about the visual cortex hierarchy in blindness (whether it parallels that of the sighted brain or is inverted) and its link to braille reading.Weaknesses:Although I have some comments and requests for clarification about the details of the methods, my main comment is that the manuscript could benefit from expanding its discussion. Specifically, I'd appreciate the authors drawing clearer theoretical conclusions about what this data suggests about the direction of information flow in the reorganized visual system in blindness, the role VWFA plays in blindness (revised from the original sighted role or similar to it?), how information arrives to the visual cortex, and what the authors' predictions would be if a parallel experiment would be carried out in sighted people (is this a multisensory recruitment or reorganization?). The data has the potential to speak to a lot of questions about the scope of brain plasticity, and that would interest broad audiences.

We thank the reviewer for the opportunity to provide clearer theoretical conclusions from our data. We elaborate on each of the points raised by the reviewer in the discussion section.

Concerning the direction of information flow in the reorganized visual system in blindness, we focus on information arrival to EVC and information flow beyond EVC.

p. 11, ll. 376-386, Discussion 4.1:

“Overall, identifying braille letter representations in widespread brain areas raises the question of how information flow is organized in the visually deprived brain. Functional connectivity studies report deprivation-driven changes of thalamo-cortical connections which could explain both arrival of information to and further flow of information beyond EVC. First, the coexistence of early thalamic connections to both S1 and V1 (Müller et al., 2019) would enable EVC to receive from different sources and at different timepoints. Second, potentially overlapping connections from both sensory cortices to other visual or parietal areas (Ioannides et al., 2013) could enable the visually deprived brain to process information in a widespread and interconnected array of brain areas. In such a network architecture, several brain areas receive and forward information at the same time. In contrast to information discretely traveling from one processing unit to the next in the sighted brain’s processing cascade, we can rather picture information flowing in a spatially and functionally more distributed and overlapping fashion.”

Regarding the role of VWFA, we propose that the functional organization of VWFA is modality-independent.

p. 10, ll. 346-348, Discussion 4.1:

“Second, we found that VWFA contains perceptual but not sensory braille letter representations. By clarifying the representational format of language representations in VWFA, our results support previous findings of the VWFA being functionally selective for letter and word stimuli in the visually deprived brain (Reich et al., 2011; Striem-Amit et al., 2012; Liu et al., 2023). Together, these findings suggest that the functional organization of the VWFA is modality-independent (Reich et al., 2011), depicting an important contribution to the ongoing debate on how visual experience shapes representations along the ventral stream (Bedny et al., 2021).” Lastly, we would like to share our thoughts about carrying out a parallel experiment in sighted people.

In general, we agree that it seems insightful to conduct a parallel, analogous experiment in sighted participants with the aim to disentangle whether the effects seen in blind participants are due to multisensory recruitment or reorganization. However, before making predictions regarding the outcome, we would have to define an analogous experiment in sighted participants that taps into the same mechanisms. This, however, is difficult to do as it is unclear what counts as analogous. For example, if we compare braille reading to reading visually presented braille dot arrays or Roman letters, we will assess visual object processing, a different mechanism from that involved in braille reading. Alternatively, if we compare braille reading to sighted participants reading embossed Roman letters haptically or ideally even reading Braille after extensive training, we still face the inherent problem that sighted participants have visual experiences and could use visual imagery strategies in these nonvisual tasks. As we cannot experimentally ensure that sighted participants do not use visual strategies to solve a task, this would always complicate drawing conclusions about the underlying processes. More specifically, we could never pinpoint whether differences between sighted and blind participants are due to measuring different mechanisms or measuring the same mechanism and unravelling underlying changes (i.e., multisensory recruitment or reorganization). Finally, apart from potential confounds due to visual imagery, considering populations of sighted readers and Braille readers as only differing with regard to their input modality and otherwise being comparable is problematic: In general, blind populations are more heterogenous than most typical samples due to various factors such as aetiologies, onset and severity (Merabet & Pascual-Leone, 2010). Even when carrying out studies in highly specific population subsamples, such as in congenitally blind braille readers, vast within-group differences remain, e.g., the quality and quantity of their braille education, as well as across braille and print readers, e.g., different passive exposure to braille versus written letters during childhood (Englebretson et al., 2023). Hence, to fully match the groups in terms of learning experience we would, for example, have to teach sighted infants braille reading in childhood and follow them up until a comparable age. This approach does not seem feasible.

p. 10, ll. 328-341, Discussion 4.1:

“We note that our findings contribute additional evidence but cannot conclusively distinguish between the competing hypotheses that visually deprived brains dynamically adjust to the environmental constraints versus that they undergo a profound cortical reorganization. Resolving this debate would require an analogous experiment in sighted people which taps into the same mechanisms as the present study. Defining a suitable control experiment is, however, difficult. Any other type of reading would likely tap into different mechanism than braille reading. Further, whenever sighted participants are asked to perform a haptic reading task, outcomes can be confounded by visual imagery driving visual cortex (Dijkstra et al., 2019). Thus, the results would remain ambiguous as to whether observed differences between the groups index different mechanisms or plastic changes in the same mechanisms. Last, matching groups of sighted readers and braille readers such that they only differ with regard to their input modality seems practically unfeasible: There are vast differences within the blind population in general, e.g., aetiologies, onset and severity, and the subsample of congenitally blind braille readers more specifically, e.g., the quality and quantity of their braille education, as well as across braille and print readers, e.g., different passive exposure to braille versus written letters during childhood (Englebretson et al., 2023; Merabet & Pascual-Leone, 2010).”

While we appreciate that the conclusions we can draw from our results are limited by our sample and defining an appropriate parallel experiment in sighted participants is difficult for the reasons discussed above, we would still like to share our speculations regarding the process underlying our result pattern. We think that our results, taken together with results of previous studies, suggest that EVC does not undergo fundamental reorganization in the case of visual deprivation. Rather, it can flexibly adjust to given processing requirements. This flexibility is not infinite; adjustments are limited by the area’s architectural and computational capacity. Importantly, we think that this claim refers to an unmasking of preexisting potential rather than multisensory recruitment.

To aid in drawing even more concrete conclusions about the flow of information, I suggest that the authors also add at least another early visual ROI to plot more clearly whether EVC's response to braille letters arrives there through an inverted cortical hierarchy, intermediate stages from VWFA, or directly, as found in the sighted brain for spoken language.

We thank the reviewer for this comment. However, EVC here consists of V1 to V3, and we already also assess V4, LOC, VWFA and LFA. Thus, we assess regions at all levels of processing from mid- over low- to high-level and cannot add a further interim ROI. Our results using this ROI set do not allow us to arbitrate between the hypotheses raised by the reviewer.

Similarly, it may be informative to look specifically at the occipital electrodes' time differences between decoding for the different parameters and their correlation to behavior.

We thank the reviewer for this suggestion. However, the spatial resolution of EEG measurements is limited, and we cannot convincingly determine the neural source of signals being recorded from specific electrodes, i.e., occipital. When we reduce the number of electrodes before analysis, we primarily see comparable qualitative trends in the data albeit with a reduction in signal-to-noise-ratio.

To illustrate, we repeated the EEG time decoding and the EEG-behavior RSA with only occipital and parieto-occipital electrodes (n=8) instead of all electrodes (n=63) and added the results to the Supplementary Material (see Supplementary Figure 3 and 4). Overall, we observe a reduction in signal-to-noise-ratio. This is not surprising given that the EEG searchlight decoding results (Figure 3b) reveal sources of the decoding signals extend beyond occipital and parieto-occipital electrodes.

In the EEG time decoding analysis, we see a comparable trend to the whole brain EEG analysis but do not find a significant difference in onsets of sensory and perceptual representation.

In the behavior-EEG RSA, we do find that the correlations between behavior and sensory representations emerge significantly earlier than correlations between behavior and perceptual representations. (N = 11, 1,000 bootstraps, one-tailed bootstrap test against zero, P< 0.001). This result is in line with the whole brain EEG analysis.

Regarding the methods, further detail on the ability to read with both hands equally and any residual vision of the participants would be helpful.

We thank the reviewer for raising this point. We assessed participants’ letter reading capabilities in a short screening task prior to the experiment. Participants read letters with both hands separately and we used the same presentation time as in the experiment. As the result showed that average performance for recognizing letters with the left hand (89%) and right hand (88%) were comparable. We did not measure continuous reading in the present study, and we did not assess further information about participants’ ability to read equally well with both hands.

While the information about the screening task was previously included in Methods section 5.3.2 EEG experiment, we now moved it into a separate section 5.3.3 Braille screening task to make the information better accessible.

p. 14, ll. 529-533, Methods 5.3.3:

“Prior to the experiment, participants completed a short screening task during which each letter of the alphabet was presented for 500ms to each hand in random order. Participants were asked to verbally report the letter they had perceived to assess their reading capabilities with both hands using the same presentation time as in the experiment. The average performance for the left hand was 89% correct (SD = 10) and for the right hand it was 88% correct (SD = 13).”

We thank the reviewer for the suggestion to include information regarding participant’s residual vision. We now added information about participants’ residual light perception to Supplementary Table 1.

**Recommendations for the authors:**

**Reviewer #1 (Recommendations For The Authors):**
(1) ROI vs Searchlight Results: Figures 2 b and c do not seem to match. The ROI results (b) should be somehow consistent with the whole brain results (c), but "perceptual" decoding in the searchlight (in green) seems localized in sensorimotor areas while for the same classification, no sensorimotor ROI is significant. can the authors clarify this difference?Similarly, perceptual decoding does not emerge in EVC with the searchlight analysis, whereas is quite strong in ROI analysis.

We agree that the results of the ROI and searchlight decoding do not show a direct match. We think that this difference is due to methodological reasons. For example, ROI decoding can be more sensitive when ROIs follow functionally relevant boundaries in the brain, in comparison to spheres used in searchlight decoding that do not. In turn, searchlight decoding may be more sensitive when information is distributed across functional boundaries that would be captured in different ROIs rather than combined, or when ROI definition is difficult (such as here in the visual system of blind participants).

However, we point out that the primary goal of our searchlight decoding was to show that no other areas beyond our hypothesized ROIs contained braille letter representations, rather than reproducing the ROI results.

Decoding accuracies are tested against chance (50% for pairwise classifications) according to methods. In the case of "sensory and perceptual" and "perceptual" classification, this is straightforward. In the case of the analysis that isolates "sensory" representations though the difference is computed between "sensory and perceptual" and "perceptual" decoding accuracies, the accuracies resulting from this difference should thus be centered around 0.Are the accuracies tested against 0 in this case? This is not specified in the methods. Furthermore, the data reported in Figure 2 and Figure 3. seem to have 0% as a baseline and the label states "decoding accuracy". Can the authors clarify whether the reported data are the difference in accuracy with an estimated empirical baseline or an expected baseline of 50%?

The reviewer is correct in stating that we tested “sensory and perceptual” and “perceptual” against chance level and the difference score “sensory” against 0 and that this information was missing in the methods section.

We now specify in the methods that we are testing the accuracies for the “sensory” analysis against 0.

p. 16, ll. 625-627, Methods 5.6:

“We conducted subject-specific braille letter classification in two ways. First, we classified between letter pairs presented to one reading hand, i.e., we trained and tested a classifier on brain data recorded during the presentation of braille stimuli to the same hand (either the right or the left hand). This yields a measure of hand-dependent braille letter information in neural measurements. We refer to this analysis as within-hand classification. Second, we classified between letter pairs presented to different hands in that we trained a classifier on brain data recorded during the presentation of stimuli to one hand (e.g., right), and tested it on data related to the other hand (e.g., left). This yields a measure of hand-independent braille letter information in neural measurements. We refer to this analysis as across-hand classification. We tested both within-hand and across-hand pairwise classification accuracies against a chance level of 50%. We also calculated a within-across hand classification score which we compared against 0.”

Regarding Figures 2 and 3, we plot the results as decoding accuracies minus chance level to standardize the y-axes for all three analyses, i.e., compare them to 0. We have corrected the y-axis labels accordingly.

In our analyses, we assumed an expected baseline of 50%. But in the response below we provide evidence that our results remain stable whether using an expected or empirical baseline.

If my understanding is correct, a potential problem persists. The different analyses may not be comparable, because in the "sensory" analysis the baseline is empirically defined, being the classification accuracies of the "perceptual" decoding, while in the other two analyses, the baseline is set at 50%. There are suggestions in the literature to derive empirically defined baselines by randomly shuffling the trial labels and repeating the classification accuracies [grootswagers 2017]. In the context of the present work, its use will make the different statistical analyses more comparable. I would thus suggest the authors define the baseline empirically for all their analyses or, given the high computational demand of this analysis, provide evidence that the results are not affected by this difference in the baseline.

We thank the reviewer for raising this point. As the reviewer correctly stated, the “sensory” analysis has an empirically defined baseline because it is a difference score while in the other two analyses the baseline is set at 50%.

To provide evidence that our results are not affected by this difference in baseline, we now re-ran the EEG time decoding. We derived null distributions from the empirical data for all three analyses, following the guidelines from Grootswagers 2017 (page 688, section “Evaluation of Classifier Performance and Group Level Statistical Testing Statistical”):

“Another popular alternative is the permutation test, which entails repeatedly shuffling the data and recomputing classifier performance on the shuffled data to obtain a null distribution, which is then compared against observed classifier performance on the original set to assess statistical significance (see, e.g., Kaiser et al., 2016; Cichy et al., 2014; Isik et al., 2014). Permutation tests are especially useful when no assumptions about the null distribution can be made (e.g., in the case of biased classifiers or unbalanced data), but they take much longer to run (e.g., repeating the analysis 10,000 times).”

Running a sign permutation test with 10,000 repetitions, we show that the results are comparable to the previously reported results based on one-sided Wilcoxon signed rank tests. We are, therefore, confident that our reported results are not affected by this difference in baseline. We now added this control analysis to the results section and supplementary material (see Supplementary Figure 5).

p. 7-8, ll. 213-215, Results 3.2:

“Importantly, the temporal dynamics of sensory and perceptual representations differed significantly. Compared to sensory representations, the significance onset of perceptual representations was delayed by 107ms (21-167ms) (N = 11, 1,000 bootstraps, one-tailed bootstrap test against zero, P = 0.012). This results pattern was consistent when defining the analysis baseline empirically (see Supplementary Figure 5).”

(2) According to the authors, perceptual rather than sensory braille letter representations identified in space are suitably formatted to guide behavior. However, they acknowledge that this finding is likely to be task-dependent because it is based on subject similarity ratings.Maybe they could use a more objective similarity measurement of Braille letters similarity?For instance, they can compare letters using Jaccard similarity (See for instance: Bottini et al. 2022).

We thank the reviewer for the opportunity to clarify. We acknowledge that our findings regarding the behavioral relevance of the identified neural representations are task-dependent. But, importantly, this is not because we use perceived similarity ratings as a measurement, but because we only use one measurement while there are infinitely many other potential tasks to assess behavior. This means that the same limitation holds when using another similarity measure like Jaccard similarity. We now clarify this in the Discussion section:

p. 12, ll. 419-420, Discussion 4.3:

“Our results clarified that perceptual rather than sensory braille letter representations identified in space are suitably formatted to guide behavior. However, we only use one specific task to assess behavior and, therefore, acknowledge that this finding is taskdependent.”

Nevertheless, we calculated Jaccard similarity based on the definition used in Bottini et. al. There are no significant correlations for the EEG-behavior or fMRI-behavior RSA when we use the Jaccard matrix and subject-specific EEG or fMRI RDMs (see Supplementary Figure 6).

This demonstrates that braille letter similarity ratings are significantly correlated with neural representations in space and time but Jaccard similarity of braille dot overlaps is not.

(3) If the primacy of perceptual similarity holds also with more objective measures of letter similarity, I think the authors should spend a few more words characterizing the results in fMRI and EEG that are rather divergent (concerning this analysis). Indeed, EEG analysis shows a significant correlation between similarity ratings and within-hand classification accuracy, although this correlation does not emerge in the "sensory" ROIs. I think these findings can be put together, hypothesizing that sensory-based similarity correlates with behavior but only in perceptual ROIs. However, why so? Can the authors provide a more mechanistic explanation? Am I missing something?

We thank the reviewer for this intriguing idea. We now speculate about how we could harmonize the results from the behavior-EEG and behavior-fMRI RSAs in the discussion section.

p. 12, ll. 438-442, Discussion 4.3:

“Similarity ratings and sensory representations as captured by EEG are correlated, and so are similarity ratings and representations in perceptual ROIs, but not sensory ROIs. This might be interpreted as suggesting a link between the sensory representations captured in EEG and the representations in perceptual ROIs. However, we do not have any evidence towards this idea. Differing signalto-noise ratios for the different ROIs and sensory versus perceptual analysis could be an alternative explanation.“

(4) In the methods they state that EEG decoding is tested against chance at each time point but these results are not reported, only latency analysis is reported. Can the authors report the significant time points of the EEG time series decoding?

We thank the reviewer for catching this inconsistency! We have now added this information to Figure 3a.

(5) In fMRI ROI definition procedure, the top 321 voxels of each anatomical ROI that had the highest functional activation were selected. The number of voxels is based on the smaller ROI, which to my understanding means that for this ROI all the voxels were selected potentially introducing noise and impacting the comparison between ROIs. Can the authors clarify which ROI was the smallest?

Thank you for the question! The smallest ROI was V4. This indeed means that for this ROI all voxels were selected. This could have led to our results being noisy in V4 but should not influence the results in other ROIs. We now added this information to the methods section. p. 15, ll. 592, Methods 5.4.4:

“The smallest mask was V4 which included 321 voxels.”

(6) Finally, the author suggests that: "Importantly, higher-level computations are not limited to the EVC in visually deprived brains. Natural sound representations 41 and language activations 53 are also located in EVC of sighted participants. This suggests that EVC, in general, has the capacity to process higher-level information 54. Thus, EVC in the visually deprived brain might not be undergoing fundamental changes in brain organization 53. This promotes a view of brain plasticity in which the cortex is capable of dynamic adjustments within pre-existing computational capacity limits 4,53-55." - The presence of a sighted control group would have strengthened this claim.

We agree with the reviewer and now discuss the limitations of our approach in the discussion section (see response to weaknesses raised by Reviewer 2 in the Public Review above).

**Reviewer #2 (Recommendations For The Authors):**
(1) Can the authors comment on the reaction time of the two reading hands? Completely ambidextrous reading is not necessarily common, so any differences in ability or response time across the hands may affect the EEG results. Alternatively, do the authors have any additional behavioral data about the participants' ability to read well with both hands?

We thank the reviewer for these questions! We did not assess reaction times and acknowledge this as a limitation. We did, however, measure accuracies and would have expected to see a speed-accuracy-trade off if reaction times would differ between hands, i.e., we would have expected lower accuracy for the hand with higher RTs. But this was not the case: our participants had comparable accuracy values when reading letters with both hands (see methods section 5.3.3 and answer to Public Review above). This measure indicated that participants recognized Braille letters presented for 500ms equally well with both index fingers.

(2) Please add information about any residual sight in the blind participants (or are they all without light perception?)

We have now added information about residual light perception in Supplementary Table 1 (see above in response to Public Review).

(3) Is active tactile exploration involved, or are the participants not moving their fingers at all over the piezo-actuators? Can the authors elaborate more on how the participants used this passive input?

We thank the reviewer for the opportunity to clarify. Our experimental setup does not involve tactile exploration or sliding motions. Instead, participants rest their index fingers on the piezo-actuators and feel the static sensation of dots pushing up against their fingertips. We assume that participants used the passive input of specific dot stimulation location on fingers to perceive a dot array which, in turn, led to the percept of a braille letter.

We now specify this information in the methods section.

p. 13, ll. 474-475, Methods 5.2:

“The modules were taped to the clothes of a participant for the fMRI experiment and on the table for the EEG and behavioral experiment. This way, participants could read in a comfortable position with their index fingers resting on the braille cells to avoid motion confounds. Importantly, our experimental setup did not involve tactile exploration or sliding motions. We instructed participants to read letters regardless of whether the pins passively stimulated their immobile right or left index finger.”

(4) I appreciated the RSA analysis, but remain curious about what the ratings were based on.Do the authors know what parameters participants used to rate for? Were these consistent across participants? That would aid in interpreting the results.

We thank the reviewer for the interest in our representational similarity analyses linking the neural representations to behavior.

We do not know which parameters participants explicitly used to rate the similarity between letters. We instructed participants to freely compare the similarity of pairs of braille letters without specifying which parameters they should use for the similarity assessment. We speculate that participants used a mixture of low-level features such as stimulation location on fingers and higher-level features such as linguistic similarity between letters. We now clarify the free comparison of braille letter pairs in the methods section:

p. 14, ll. 538-539, Methods 5.3.4:

“Each pair of letters was presented once, and participants compared them with the same finger. We instructed participants to freely compare the similarity of pairs of Braille letters without specifying which parameters they should use for the similarity assessment. The rating was without time constraints, meaning participants decided when they rated the stimuli. Participants were asked to verbally rate the similarity of each pair of braille letters on a scale from 1 = very similar to 7 = very different and the experimenter noted down their responses.”

(5) Can the authors provide confusion matrices for the decoding analyses in the supplementary materials? This could be informative in understanding what pairs of letters are most discernable and where.

We have added confusion matrices for within- and between-hand decoding for all ROIs and for the time points 100ms, 200ms, 300ms and 400ms to the Supplementary Material (see Supplementary Figures 7-10).

(6) Was slice time correction done for the fMRI data? This is not reported.

We now added this information to the methods section - our fMRI preprocessing pipeline did not include slice timing correction.

p. 14, ll. 554, Methods 5.4.2:

“We did not apply high or low-pass temporal filters and did not perform slice time correction.”